# What has women's reproductive health decision-making capacity and other factors got to do with pregnancy termination in sub-Saharan Africa? evidence from 27 cross-sectional surveys

Abdul-Aziz Seidu[1,2], Bright Opoku Ahinkorah[3], Edward Kwabena Ameyaw[3], Amu Hubert[1,4], Wonder Agbemavi[1], Ebenezer Kwesi Armah-Ansah[1], Eugene Budu[1], Francis Sambah[5]*, Vivian Tackie[5,6]

1 Department of Population and Health, University of Cape Coast, Cape Coast, Ghana, 2 College of Public Health, Medical and Veterinary Sciences, James Cook University, Townsville, Queensland, Australia, 3 The Australian Centre for Public and Population Health Research (ACPPHR), Faculty of Health, University of Technology Sydney, Sydney, Australia, 4 Department of Population and Behavioural Sciences, School of Public Health, University of Health and Allied Sciences, Hohoe, Ghana, 5 Department of Health, Physical Education and Recreation, University of Cape Coast, Cape Coast, Ghana, 6 Department of Public Health, School of Nursing and Midwifery, University of Health and Allied Sciences, Ho, Ghana

* sambahfrancis80@gmail.com

## Abstract

### Introduction

Pregnancy termination is one of the key issues that require urgent attention in achieving the third Sustainable Development Goal (SDG) of ensuring healthy lives and promoting well-being for all at all ages. The reproductive health decision-making (RHDM) capacity of women plays a key role in their reproductive health outcomes, including pregnancy termination. Based on this premise, we examined RHDM capacity and pregnancy termination among women of reproductive age in sub-Saharan Africa (SSA).

### Materials and methods

We pooled data from the women's files of the most recent Demographic and Health Surveys (DHS) of 27 countries in SSA, which are part of the DHS programme. The total sample was 240,489 women aged 15 to 49. We calculated the overall prevalence of pregnancy termination in the 27 countries as well as the prevalence in each individual country. We also examined the association between RHDM capacity, socio-demographic characteristics and pregnancy termination. RHDM was generated from two variables: decision-making on sexual intercourse and decision-making on condom use. Binary logistic regression analysis was conducted and presented as Crude Odds Ratios (COR) and Adjusted Odds Ratios (AOR) with their corresponding 95% confidence intervals (CI). Statistical significance was declared p<0.05.

**Data Availability Statement:** The data set is available to the public at https://dhsprogram.com/data/available-datasets.cfm.

**Funding:** The authors received no specific funding for this work.

**Competing interests:** The authors have declared that no competing interest exist.

## Results

The prevalence of pregnancy termination ranged from 7.5% in Benin to 39.5% in Gabon with an average of 16.5%. Women who were capable of taking reproductive health decisions had higher odds of terminating a pregnancy than those who were incapable (AOR = 1.20, 95% CI = 1.17–1.24). We also found that women aged 45–49 (AOR = 5.54, 95% CI = 5.11–6.01), women with primary level of education (AOR = 1.14, 95% CI = 1.20–1.17), those cohabiting (AOR = 1.08, 95% CI = 1.04–1.11), those in the richest wealth quintile (AOR = 1.06, 95% CI = 1.02–1.11) and women employed in the services sector (AOR = 1.35, 95% CI = 1.27–1.44) were more likely to terminate pregnancies. Relatedly, women who did not intend to use contraceptive (AOR = 1.47, 95% CI = 1.39–1.56), those who knew only folkloric contraceptive method (AOR = 1.25, 95% CI = 1.18–1.32), women who watched television almost every day (AOR = 1.16, 95% CI = 1.20–1.24) and those who listened to radio almost every day (AOR = 1.11, 95% CI = 1.04–1.18) had higher odds of terminating a pregnancy. However, women with four or more children had the lowest odds (AOR = 0.5, 95% CI = 0.54–0.60) of terminating a pregnancy.

## Conclusion

We found that women who are capable of taking reproductive health decisions are more likely to terminate pregnancies. Our findings also suggest that age, level of education, contraceptive use and intention, place of residence, and parity are associated with pregnancy termination. Our findings call for the implementation of policies or the strengthening of existing ones to empower women about RHDM capacity. Such empowerment could have a positive impact on their uptake of safe abortions. Achieving this will not only accelerate progress towards the achievement of maternal health-related SDGs but would also immensely reduce the number of women who die as a result of pregnancy termination in SSA.

## Introduction

In 2015, the United Nations launched the 2030 Agenda for the Sustainable Development Goals (SDGs) [1]. Goal Three seeks to ensure healthy lives and promote well-being for all at all ages. This goal highlights the need to reduce maternal mortality and to improve reproductive health [2]. Reproductive health refers to a state of complete physical, mental and social well-being, and not merely the absence of disease or infirmity in all matters relating to the reproductive system and to its functions and processes [3]. The implication is that individuals can attain a safe and satisfying sexual life, procreate and freely decide if, when and how often to do so [3]. Reproductive health is also considered a central constituent of a person's general health status and an integral contributory factor to quality of life [3].

Pregnancy termination has been found as one of the key issues that needs to be addressed to achieve SDG Three by 2030 [4]. This is because globally about 830 women die from pregnancy and childbirth related causes annually and 99% of such deaths occur in Low- and Middle-Income Countries (LMICs) [5, 6]. Unsafe pregnancy termination is a major contributory factor and remains a pandemic and serious public health issue worldwide [7, 8]. Worldwide about 97% of all unsafe pregnancies terminated between 2010 and 2014 occurred in LMICs [9]. In Africa, over 4 million unsafe abortions are carried out yearly; mostly among the poor, rural, and

young women lacking information on the availability of safe abortion care [10]. About 99% of all pregnancy terminations carried out in Africa are unsafe, and the risk of maternal death from an unsafe abortion is 1 in every 150 procedures which is the highest in the world [10].

Pregnancy termination also known as abortion, may occur either spontaneously or intentionally. The latter also known as induced abortion, may be either safe or unsafe [5]. The World Health Organization (WHO) [6] defines unsafe abortion as a procedure for terminating a pregnancy performed by persons lacking the necessary skills or in an environment, not in conformity with minimal medical standards, or both.

Pregnancy termination (especially the unsafe) can have serious health consequences and cause complications such as haemorrhage, sepsis and uterine perforation [7, 11]. Unsafe abortion also has undesirable consequences beyond its immediate effects on women's health. For example, complications propelled by unsafe abortion can lead to a reduction in women's productivity, increase the economic burden on poor families, and bring about substantial costs to already struggling public health systems [12]. In sub-Saharan Africa (SSA), pregnancy termination is more common, done clandestinely and contributes substantially to maternal mortality [13–15].

To achieve SDG Three, it is important to enhance universal access to sexual and reproductive health services that guarantee the health needs and aspirations of women of reproductive age [16]. However, in many societies, especially in SSA, the status of women does not offer them the capacity to make decisions relating to many aspects of their lives [17]. The decision-making ability of a woman regarding her reproductive health will be efficiently achieved depending on her capacity to afford her needs [18]. In LMICs, especially SSA, myriad cultural and socio-economic factors affect the ability of women to make decisions regarding their reproductive health [19]. Cultural traditions and beliefs in the sub-region, for instance, endorse the hierarchical role of men in sexual relationships and especially marriage [20], which makes it difficult for women to be the key deciders of their reproductive health.

A study by Seidu et al. [21] indicates that the RHDM capacity of women plays a key role in their reproductive health outcomes, including pregnancy termination. Specifically, the authors concluded that women with RHDM capacity are more likely to terminate pregnancies. Although this study does not clearly specify if women with RHDM capacity undergo safe or unsafe pregnancy termination, they linked RHDM to empowerment, which gives women the capacity to have control over their reproductive health and can access pregnancy termination in health facilities [21], where safe pregnancy terminations mostly occur.

In SSA, some studies have been conduct at country-levels on women's reproductive health decision making [20, 22–25] and pregnancy termination [26–28]. At the sub-regional level (SSA), there have been studies on women's reproductive health decision making [18, 20, 29–31] and pregnancy termination [10, 13, 32]. All these studies either focused mainly on the determinants of RHDM or predictors of pregnancy termination. Hence, the link between RHDM and pregnancy has not been established. The study that has established the link between RHDM and pregnancy is the study by Seidu et al. [21]. However, their study was a one-country study and does not provide a holistic understanding of the situation in a sub-regional context like SSA. We, therefore, sought to fill this gap in the literature by examining the RHDM capacity and pregnancy termination among women of reproductive age in 27 SSA countries using data from the most recent Demographic and Health Surveys (DHS) of the respective countries.

## Materials and methods

### Data source

Our study used pooled data from DHS conducted from January 1, 2010 to December 31, 2016 in 27 SSA countries (see Table 1). The 27 countries were included in the study because they

**Table 1. Prevalence of pregnancy termination among women in SSA.**

| Country | Weighted n = 240,489 | Weighted % | Pregnancy Terminated | |
|---|---|---|---|---|
| | | | No (%) | Yes (%) |
| Angola | 50,751 | 21.1 | 87.4 | 12.6 |
| Burkina Faso | 41,213 | 17.1 | 85.7 | 14.3 |
| Benin | 34,807 | 14.5 | 92.5 | 7.5 |
| Burundi | 3,232 | 1.3 | 80.5 | 19.5 |
| Congo DR | 3,979 | 1.7 | 81.5 | 18.5 |
| Congo | 19,328 | 8.0 | 61.2 | 38.8 |
| Côte d'Ivoire | 19,076 | 7.9 | 79.4 | 20.6 |
| Cameroon | 13,615 | 5.7 | 70.4 | 29.6 |
| Ethiopia | 3,345 | 1.4 | 89.4 | 10.7 |
| Gabon | 1,367 | 0.6 | 60.5 | 39.5 |
| Ghana | 1,764 | 0.7 | 73.7 | 26.4 |
| The Gambia | 2,190 | 0.9 | 87.2 | 12.8 |
| Guinea | 2,209 | 0.9 | 85.7 | 14.4 |
| Liberia | 1,761 | 0.7 | 77.3 | 22.7 |
| Lesotho | 489 | 0.2 | 84.3 | 15.8 |
| Mali | 2,488 | 1.0 | 90.8 | 9.2 |
| Malawi | 5,304 | 2.2 | 87.5 | 12.5 |
| Nigeria | 9,014 | 3.8 | 86.7 | 13.3 |
| Namibia | 1,015 | 0.4 | 87.5 | 12.5 |
| Rwanda | 2,275 | 1.0 | 80.6 | 19.4 |
| Sierra Leone | 3,533. | 1.5 | 89.3 | 10.7 |
| Senegal | 3,363 | 1.4 | 78.9 | 21.2 |
| Chad | 1,450 | 0.6 | 88.4 | 11.7 |
| Togo | 2,065 | 0.9 | 84.7 | 15.3 |
| Uganda | 3,709 | 1.5 | 76.8 | 23.2 |
| Zambia | 3,193 | 1.3 | 86.0 | 14.0 |
| Zimbabwe | 3,953 | 1.6 | 84.1 | 15.9 |
| Total | 240,489 | 100.0 | 83.5 | 16.5 |

had recent DHS data and had all the variables of interest for this study. The DHS is a nation-wide survey collected every five years across LMICs. The survey is representative of each of the countries and targets core maternal and child health indicators such as unintended pregnancy, contraceptive use, skilled birth attendance, immunisation among under-five children and inti-mate partner violence. Women's files were used for our study and these files possess the responses by women aged 15 to 49. For this study, a sample size of 240,489 women who had complete information on RHDM were included. Thus, women who were either married or cohabiting (living with a partner) were included. Details of the DHS methodology has been extensively described elsewhere [33]. We followed the 'Strengthening the Reporting of Obser-vational Studies in Epidemiology' (STROBE) statement in conducting this study and writing the manuscript.

## Definition of variables

**Dependent variable.** The outcome variable employed in this study was "pregnancy termi-nation". It was derived from the question "have you ever had a pregnancy terminated?" and responses were coded as 0 = "No" and 1 = "Yes".

**Explanatory variables.** In all, fifteen explanatory variables were considered, with RHDM capacity as the main explanatory variable. RHDM was derived from two variables, decision-making on sexual intercourse and decision-making on condom use. For decision making on sexual intercourse, women were asked if they can refuse their partner sex while for decision-making on condom use, women were asked if they can ask their partners to use condoms during sexual activity. Like previous studies on RHDM (see 18, 21, 22), the original response category of these variables (1 = yes, 2 = no and 3 = don't know/ not sure) were categorized as 0 = "no and don't know" and 1 = "yes" in the present study. RHDM capacity, was then generated by combining the decision-making on sexual intercourse and the decision-making on condom use variables. This was categorized as 0 = "not capable" and 1 = at least capable of taking 1 decision (capable). Hence, women who gave 'Yes' responses to questions on both sexual intercourse and the decision-making on condom use were considered as having RHDM capacity [18, 21, 22]. Apart from RHDM capacity, the other explanatory variables were country, age of respondent, educational level, marital status, wealth status, working status, religion, place of residence, parity, contraceptive use and intention, knowledge on contraceptive, exposure to newspapers, exposure to television, and exposure to radio. These explanatory variables were included in this study primarily based on the conclusions drawn on them from previous studies [21, 34, 35] to be associated with pregnancy termination. Four of these variables were recoded to make them meaningful for analysis and interpretation. Marital status was recoded into 'never married (0)', 'married (1)', 'cohabiting (2)'. Working status was captured as 'not working (0)', 'managerial (1)', 'clerical (2)', 'sales (3)', 'agricultural (4)', 'household (5)', 'services (6)' and 'manual (7)'. We recoded parity as 'zero birth (0)', 'one birth (1)', 'two births (2)', 'three births (3)', and four or more births (4)'. Finally, religion was recoded as 'Christianity (1)', 'Islam (2)', ' no religion (3)', and 'other (4)'.

## Statistical analyses

The analysis began with the computation of pregnancy termination for each of the 27 SSA countries. We then appended the datasets and this generated a total sample of 240,489. After appending, we calculated the overall prevalence of pregnancy termination across the explanatory variables. Two Binary Logistic Regression analyses were built. The first model (Model I) was a bivariate model and included the main explanatory variable (RHDM capacity) and the outcome variable (pregnancy termination) only. In Model II (Multivariable), we adjusted for the effect of country and the other explanatory variables to ascertain how these variables interact with RHDM capacity to influence pregnancy termination. The choice of reference categories for these explanatory variables was informed by previous studies [21, 34, 35]. Binary logistic regression was employed because our outcome variable (pregnancy termination) had a dichotomous outcome. The results were presented as Crude Odds Ratios (COR) and Adjusted Odds Ratios (AOR) with their corresponding 95% confidence intervals (CIs) signifying level of precision. The Hosmer-Lemeshow test was adopted to test the appropriateness of the model specification. Multicollinearity was checked with Variance Inflation Factors (VIF). We had a mean VIF of 1.55 confirming that there was no evidence of multicollinearity. Weighted frequencies were generated while the survey command in STATA was used to account for the complex nature of the data in the regression analyses to produce unbiased robust standard errors. All analyses were carried out with STATA version 14.2 for MacOS. The statistical significance level was set at $p < 0.05$.

## Ethical clearance

The DHS obtain ethical clearance from the Ethics Committee of ORC Macro Inc. as well as Ethics Boards of partner organizations of the various countries such as the Ministries of

Health. During each of the surveys, either written or verbal consent was provided by the women. Since the data was not collected by the authors of this manuscript, we sought permission from the website of MEASURE DHS and access to the data was provided after our intent for the request was assessed and approved on 27th January, 2019. Permission to use the dataset was obtained from MEASURE DHS. The data set is available to the public at https://dhsprogram.com/data/available-datasets.cfm.

## Results

### Prevalence of pregnancy termination among women in SSA

Table 1 presents the results of the prevalence of pregnancy termination among women in SSA. Overall, the prevalence of pregnancy termination among the respondents was 16.5%. It ranged from 7.5% in Benin to 39.5% in Gabon.

### RHDM capacity, socio-demographic characteristics and pregnancy termination among women in SSA

Table 2 presents the results of the study on RHDM capacity, socio-demographic characteristics and pregnancy termination among women in SSA. It was found that 18.0% of women who could take reproductive health decision, had ever had a pregnancy terminated. The prevalence of pregnancy termination was highest among respondents aged 45–49 (23.4%). It was also high among those who were cohabiting (20.1%), those in urban areas (18.0%), those with a secondary level of education (18.9%), those in the richest wealth quintile (18.3%) and those with parity 4 and above (18.5%). The prevalence was also higher among respondents whose occupation was managerial (20.2%), those who belong to 'other' religious sects, and those who use traditional contraception. Besides, pregnancy termination was more prevalent (17.2%) among those who knew modern methods of contraception. In terms of media exposure, women who read a newspaper (25.0%), watched television (24.7%) and who listened to radio almost every day (21.4%) had the highest prevalence of pregnancy termination. The chi-square analysis showed that all the explanatory variables were associated with pregnancy termination (p<0.001).

### Logistic regression analysis results on RHDM capacity and pregnancy termination among women in SSA

Table 3 presents logistic regression analyses on RHDM capacity and pregnancy termination in SSA. There was a significant relationship between pregnancy termination and RHDM capacity. We found that women who were capable of taking reproductive health decisions had higher odds of terminating a pregnancy compared to those who were incapable (AOR = 1.20, 95% CI = 1.17–1.24). In terms of country, women in Gabon (AOR = 3.54, 95% CI = 3.18–3.93) and Congo (AOR = 3.54, 95% CI = 3.188–3.92) had higher odds of terminating a pregnancy compared to those in Angola. In terms of age and pregnancy termination, the highest was among those aged 45–49 (AOR = 5.54, 95% CI = 5.11–6.01). Women with primary level of education (AOR = 1.14, 95% CI = 1.20–1.17) and those cohabiting (AOR = 1.08, 95% CI = 1.04–1.11), those in richest wealth quintile (AOR = 1.06, 95% CI = 1.02–1.11) and women in services (AOR = 1.35, 95% CI = 1.27–1.44) had higher odds of terminating pregnancy. Women who did not intend to use contraception (AOR = 1.47, 95% CI = 1.39–1.56) and knew only folkloric method (AOR = 1.25, 95% CI = 1.18–1.32) had higher odds of terminating a pregnancy compared to those who knew no method. Women with parity four or more had lower odds of terminating a pregnancy compared to nulliparous women. With media

**Table 2. RHDM capacity, socio-demographic characteristics and pregnancy termination among women in SSA (n = 240,489).**

| Variables | Weighted n | Weighted % | Pregnancy Terminated | | P-value |
|---|---|---|---|---|---|
| | | | No (%) | Yes (%) | |
| **RDM Capacity** | | | | | p<0.001 |
| Incapable | 76,687 | 31.9 | 87.1 | 12.9 | |
| Capable | 163,802 | 68.1 | 82.0 | 18.0 | |
| **Age** | | | | | p<0.001 |
| 15–19 | 12,705 | 5.3 | 93.0 | 7.0 | |
| 20–24 | 42,124 | 17.5 | 89.2 | 10.9 | |
| 25–29 | 53,382 | 22.2 | 85.7 | 14.3 | |
| 30–34 | 45,833 | 19.1 | 82.9 | 17.1 | |
| 35–39 | 38,001 | 15.8 | 79.8 | 20.2 | |
| 40–44 | 28,182 | 11.7 | 77.2 | 22.8 | |
| 45–49 | 20,261 | 8.4 | 76.6 | 23.4 | |
| **Marital status** | | | | | p<0.001 |
| Married | 156,011 | 64.9 | 84.5 | 15.6 | |
| Cohabiting | 84,478 | 35.1 | 79.9 | 20.1 | |
| **Place of Residence** | | | | | p<0.001 |
| Urban | 100,651 | 41.9 | 82.0 | 18.0 | |
| Rural | 139,838 | 58.2 | 84.3 | 15.8 | |
| **Educational level** | | | | | p<0.001 |
| No education | 115,631 | 48.1 | 86.0 | 14.0 | |
| Primary | 65,280 | 27.1 | 82.1 | 17.9 | |
| Secondary | 53,151 | 22.1 | 81.1 | 18.9 | |
| Higher | 6,427 | 2.7 | 81.7 | 18.4 | |
| **Wealth status** | | | | | p<0.001 |
| Poorest | 47,751 | 19.9 | 84.2 | 15.8 | |
| Poorer | 49,603 | 20.6 | 84.0 | 16.0 | |
| Middle | 47,894 | 19.9 | 84.2 | 15.9 | |
| Richer | 48,859 | 20.3 | 83.3 | 16.7 | |
| Richest | 46,382 | 19.3 | 81.7 | 18.3 | |
| **Parity** | | | | | p<0.001 |
| Zero birth | 4,238 | 1.8 | 84.4 | 15.6 | |
| One birth | 35,738 | 14.9 | 86.9 | 13.1 | |
| Two births | 41,452 | 17.2 | 85.3 | 14.7 | |
| Three births | 38,521 | 16.0 | 84.5 | 15.5 | |
| Four or more births | 120,540 | 50.1 | 81.5 | 18.5 | |
| **Occupation** | | | | | p<0.001 |
| Not working | 61,165 | 25.4 | 87.3 | 12.7 | |
| Managerial | 4,995 | 2.1 | 79.8 | 20.2 | |
| Clerical | 39,692 | 16.5 | 85.8 | 14.2 | |
| Sales | 46,108 | 19.2 | 80.6 | 19.4 | |
| Agricultural | 63,718 | 26.5 | 82.3 | 17.7 | |
| Services | 11,387 | 4.7 | 81.0 | 19.0 | |
| Manual | 13,425 | 5.6 | 83.0 | 17.0 | |
| **Religion** | | | | | p<0.001 |
| Christianity | 140,408 | 58.4 | 82.2 | 17.8 | |
| Islam | 70,519 | 29.3 | 85.9 | 14.1 | |
| No region | 9,436 | 3.9 | 82.4 | 17.6 | |

(*Continued*)

**Table 2.** (*Continued*)

| Variables | Weighted n | Weighted % | Pregnancy Terminated | | P-value |
|---|---|---|---|---|---|
| | | | No (%) | Yes (%) | |
| Other | 20,125 | 8.4 | 81.5 | 18.6 | |
| **Intention to use contraceptive** | | | | | p<0.001 |
| Using modern | 40,270 | 16.7 | 84.3 | 15.7 | |
| Using traditional | 12,301 | 5.1 | 74.4 | 25.7 | |
| Non-user intends to use later | 74,878 | 31.1 | 82.9 | 17.1 | |
| Does not intend to use | 113,040 | 47.0 | 84.3 | 15.7 | |
| **Knowledge on contraceptives** | | | | | p<0.001 |
| Knows no method | 22,951 | 9.5 | 91.1 | 8.9 | |
| knows only folkloric method | 575 | 0.2 | 91.3 | 8.8 | |
| Knows traditional method | 1,399 | 0.6 | 87.3 | 12.7 | |
| Knows modern method | 215,564 | 89.6 | 82.9 | 17.2 | |
| **Frequency of Reading newspaper/Magazine** | | | | | p<0.001 |
| Not at all | 205,951 | 85.6 | 84.1 | 15.9 | |
| Less than once a week | 12,874 | 5.4 | 80.2 | 19.8 | |
| At least once a week | 18,554 | 7.7 | 81.4 | 18.6 | |
| Almost every day | 3,110 | 1.3 | 75.0 | 25.0 | |
| **Frequency of Watching Television** | | | | | p<0.001 |
| Not at all | 131,993 | 54.9 | 84.8 | 15.2 | |
| Less than once a week | 23,084 | 9.6 | 82.7 | 17.3 | |
| At least once a week | 50,347 | 20.9 | 82.4 | 17.6 | |
| Almost every day | 35,065 | 14.6 | 75.3 | 24.7 | |
| **Frequency of Listening to Radio** | | | | | p<0.001 |
| Not at all | 101,240 | 42.1 | 85.1 | 14.9 | |
| Less than once a week | 38,132 | 15.9 | 82.6 | 17.4 | |
| At least once a week | 81,030 | 33.7 | 82.8 | 17.2 | |
| Almost every day | 20,087 | 8.4 | 78.6 | 21.4 | |

* p-values are for Chi-square test.

**Table 3. Logistic regression analysis on RHDM capacity and pregnancy termination among women in SSA.**

| Variables | Model 1 COR (95% CI) | Model 2 AOR (95% CI) |
|---|---|---|
| **RHDM Capacity** | | |
| Incapable | Ref | Ref |
| Capable | 1.48***[1.45–1.52] | 1.20***[1.17–1.24] |
| **Age** | | |
| 15–19 | | Ref |
| 20–24 | | 1.74***[1.62–1.87] |
| 25–29 | | 2.63***[2.45–2.83] |
| 30–34 | | 3.46***[3.21–3.73] |
| 35–39 | | 4.36***[4.04–4.71] |
| 40–44 | | 5.22***[4.82–5.64] |
| 45–49 | | 5.54***[5.11–6.01] |
| **Educational level** | | |
| No education | | Ref |
| Primary | | 1.14***[1.10–1.17] |

(*Continued*)

**Table 3.** (Continued)

| Variables | Model 1 COR (95% CI) | Model 2 AOR (95% CI) |
|---|---|---|
| Secondary | | 1.08***[1.04–1.13] |
| Higher | | 0.94[0.87–1.02] |
| **Marital status** | | |
| Married | | Ref |
| Cohabiting | | 1.08***[1.04–1.11] |
| **Religion** | | |
| Christianity | | Ref |
| Islam | | 0.97[0.94–1.00] |
| No region | | 1.03[0.95–1.11] |
| Other | | 1.03[0.98–1.09] |
| **Employment** | | |
| Not working | | Ref |
| Managerial | | 1.17***[1.09–1.25] |
| Clerical | | 1.21***[1.11–1.32] |
| Sales | | 1.28***[1.24–1.33] |
| Agricultural | | 1.19***[1.15–1.24] |
| Services | | 1.35***[1.27–1.44] |
| Manual | | 1.29***[1.23–1.36] |
| **Parity** | | |
| Zero birth | | Ref |
| One birth | | 0.72***[0.68–0.77] |
| Two births | | 0.66***[0.62–0.70] |
| Three births | | 0.60***[0.57–0.64] |
| Four or more births | | 0.57***[0.54–0.60] |
| **Intention to use contraception** | | |
| Using modern method | | Ref |
| Using traditional method | | 0.97[0.72–1.32] |
| Non-user intends to use later | | 1.12 [0.93–1.34] |
| Does not intend to use | | 1.47***[1.39–1.56] |
| **Knowledge on contraceptives** | | |
| Knows no method | | Ref |
| Knows only folkloric method | | 1.25***[1.18–1.32] |
| Knows traditional method | | 1.21***[1.17–1.25] |
| Knows modern method | | 1.08***[1.05–1.12] |
| **Frequency of reading newspaper/Magazine** | | |
| Not at all | | Ref |
| Less than once a week | | 1.04[1.00–1.09] |
| At least once a week | | 0.97[0.93–1.02] |
| Almost every day | | 0.95[0.83–1.08] |
| **Frequency of watching Television** | | |
| Not at all | | Ref |
| Less than once a week | | 1.01[0.97–1.05] |
| At least once a week | | 1.04*[1.01–1.08] |
| Almost every day | | 1.16***[1.10–1.24] |
| **Frequency of listening to radio** | | |
| Not at all | | Ref |
| Less than once a week | | 1.13***[1.10–1.17] |

(*Continued*)

**Table 3.** (Continued)

| Variables | Model 1 COR (95% CI) | Model 2 AOR (95% CI) |
|---|---|---|
| At least once a week | | 1.10***[1.07–1.13] |
| Almost every day | | 1.11**[1.04–1.18] |
| **Wealth status** | | |
| Poorest | | Ref |
| Poorer | | 1.03[0.99–1.07] |
| Middle | | 1.00[0.96–1.04] |
| Richer | | 1.01[0.97–1.06] |
| Richest | | 1.06*[1.02–1.11] |
| **Place of residence** | | |
| Urban | | Ref |
| Rural | | 1.01[0.98–1.04] |
| **Country** | | |
| Angola | | Ref |
| Burkina Faso | | 1.13* [1.01–1.26] |
| Benin | | 0.54***[0.48–0.61] |
| Burundi | | 1.45***[1.31–1.61] |
| Congo DR | | 1.53***[1.38–1.69] |
| Congo | | 3.54***[3.19–3.92] |
| Côte d'Ivoire | | 1.73***[1.55–1.94] |
| Cameroon | | 2.58***[2.31–2.88] |
| Ethiopia | | 0.91[0.81–1.02] |
| Gabon | | 3.54***[3.18–3.93] |
| Ghana | | 1.90***[1.70–2.12] |
| Gambia | | 1.08[0.95–1.220] |
| Guinea | | 1.23***[1.09–1.38] |
| Liberia | | 1.82***[1.64–2.03] |
| Lesotho | | 0.93[0.78–1.12] |
| Mali | | 0.87*[0.77–0.99] |
| Malawi | | 0.96[0.86–1.06] |
| Nigeria | | 0.99[0.90–1.10] |
| Namibia | | 0.74***[0.64–0.85] |
| Rwanda | | 1.25***[1.11–1.39] |
| Sierra Leone | | 0.79***[0.71–0.89] |
| Senegal | | 2.10***[1.89–2.34] |
| Chad | | 1.28***[1.12–1.46] |
| Togo | | 1.00 [0.89–1.12] |
| Uganda | | 1.88***[1.69–2.08] |
| Zambia | | 1.01[0.91–1.12] |
| Zimbabwe | | 1.22***[1.10–1.36] |

*$p < 0.05$

** $p < 0.01$

*** $p < 0.001$; Ref: Reference; COR = Crude Odds Ratio; AOR = Adjusted Odds Ratio.

exposure, we realized that women who watched Television (AOR = 1.16, 95% CI = 1.20–1.24) and listened to radio almost every day (AOR = 1.11, 95% CI = 1.04–1.18) had higher odds of terminating a pregnancy compared to those who do not watch Television nor listen to radio at all.

## Discussion

In this study, we examined the influence of RHDM capacity on pregnancy termination among women in SSA. We also looked at how socio-demographic characteristics interact with RHDM capacity to predict pregnancy termination among women. Our results showed that the odds of pregnancy termination were high among women who had the capacity to make reproductive health decision. The findings confirm a previous study by Seidu et al. [21] that women with the capacity to make reproductive health decisions are more likely to terminate pregnancies compared to those who are incapable of making reproductive health decisions. Prata, Tavrow and Upadhyay [36] explained the relationship between RHDM and pregnancy termination. According to the authors, women's decision-making is essential in their reproductive health choices, including pregnancy termination because the choices they make concerning their reproductive health play significant roles in having a better life. Similarly, Uberoi and de Bruyn [37] argued that reproductive rights are central to women's self-determination and contribute significantly to making essential reproductive health decisions in their lives. Although the reason and termination procedures were not reported, women who have the capacity to make decisions concerning their reproductive health may require little or no approval from their partners when they need to terminate a pregnancy. Again, such women may be empowered to deal with constraints from their partners on matters relating to pregnancy termination. Even though RHDM capacity is essential for improving women's health [18], it's effect on pregnancy termination calls for the need to promote women's RHDM capacity while at the same time advocating strategies to achieve safe and medically sanctioned pregnancy termination.

The likelihood of pregnancy termination was found to be high in Gabon and Congo. The findings confirm the findings of Chae et al. [38], who identified Gabon and Congo as countries with high prevalence of pregnancy termination in SSA. The possible reason for the high prevalence of pregnancy termination in Gabon and Congo could be explained by the prevalence of unintended pregnancies and contraceptives in both countries. For instance, a recent study by Ameyaw et al. [23] found the prevalence of unintended in Gabon and Congo to be 43.7% and 37.1% respectively. Similarly, Gabon and Congo have been identified among African countries with contraceptive prevalence levels below 25% [39]. Both Gabon and Congo have been identified among countries in the world where pregnancy termination is illegal [40]. This implies that there is a likelihood of high unsafe termination of pregnancies in these countries. The high prevalence of pregnancy termination in Gabon and Congo calls for the need to improve reproductive health for women in both countries and empower women to make the right decisions on their reproductive health.

We also found that the likelihood of terminating a pregnancy was high among older women compared to younger women. Our findings corroborate the findings obtained in previous studies in SSA [21, 34, 41], where pregnancy termination was found to be high among older women. The possible reason for our finding is that as women age, there is a tendency of thinking that having an additional child may not be essential, especially when they have had their desired number of children. Such women will, therefore, have no much issues considering pregnancy termination when an unintended pregnancy occurs. Another possible reason for our finding is that older women are at higher risk of experiencing risks related to pregnancy [42], which may call for an abortion. For instance, Jolly et al. [43] identified gestational diabetes mellitus as significantly common in the older age groups. Other studies have also found the risk of negative pregnancy outcomes among older women [44–47]. Ageing reproductive system and an ageing body have been identified as some of the factors that increase pregnancy-related risks in older women [48].

The likelihood of pregnancy termination was found to be high among women with primary and secondary education compared to women with no formal education. Our findings confirm the findings of Klutsey and Ankomah [14] and that of Lean et al. [49], who found that educated women have a higher likelihood of induced abortion. The findings contradict the findings of Andersen et al. [50] and Dickson et al. [34], who found that the likelihood of pregnancy termination is high among women with no formal education. These authors explained that women with no education are less likely to use contraceptive to avoid unintended pregnancies that are likely to result in termination. However, we maintain that educated women may have pregnancies that could interfere with their education and hence may decide to terminate those pregnancies.

Richest women and women who were working had the highest odds of terminating pregnancy. Our findings support the findings of previous studies [5, 51, 52]. Our finding is plausible when considered in the perspective of the link between financial empowerment and access to abortion/termination services. Safe abortion services are not easily affordable in most SSA countries due to limited legal facilities and practitioners to provide these services [12]. Hence, working women and richest women may be able to access abortion services due to their financial empowerment. Additionally, women who work and the rich women may consider pregnancy as a burden on their employment and their quest for high productivity and income. Therefore, when they feel that they have already had their desired number of children, they may not hesitate to terminate an unintended pregnancy.

Our study identified high prevalence of pregnancy termination among cohabiting women. Our findings confirm the findings of DaVanzo and Rahman [53] and Andersen et al. [54] who also found that unmarried women are more likely to terminate pregnancy compared to married women. In general, never-married women tend to be more comfortable and supportive of abortion. As such, women who are cohabiting may not want to give birth outside marriage to escape ridicule from society. Moreover, unmarried women are more likely to indicate that they will support a friend who needed an abortion or feel confident that they could help a friend obtain the service [54]. The high prevalence of pregnancy termination among cohabiting women can also be linked to the high prevalence of unintended pregnancies among never married women [23, 55, 56].

We also found that women with no children had higher odds of terminating pregnancy as compared to women with parity 1 and above. Previous studies in Ghana [21] and Mozambique [34] have reported the same. Parker et al. [57] explained this by linking pregnancy to preeclampsia which is estimated to affect 4%–8% of first pregnancies. They explained that women who suffer from preeclampsia and other pregnancy related complications may experience situations that may require termination of pregnancy. The likelihood of pregnancy termination was also found to be high among women who knew about contraceptives and those that had no intention to use contraceptives. Although knowledge of contraceptives could reduce the risk of unintended pregnancies, which consequently reduces pregnancy termination, we hold that pregnancy termination will be high when knowledge of contraceptives is not accompanied by intention to use contraceptives. It is, therefore, not surprising that in our study, women with high knowledge of contraceptives and those with no intention to use contraceptives had high prevalence of pregnancy termination.

We found that women who watched television and those who listened to radio were respectively more likely to terminate pregnancies compared to those who did not. The findings thus, reinforce the role of the media in informing the reproductive health decision making of women. Congruent to previous findings by Dickson et al. [34] in Ghana and Andersen et al. [58] in India, we assert that women who had access to television and radio were probably more likely to have obtained information about abortion services and where they are offered

and hence found it easier getting pregnancies terminated compared to those who did not watch television or listen to radio.

## Strengths and limitations

This is a multi-country study using comparable datasets to investigate RHDM and pregnancy termination in SSA. With the large dataset, validity measures underlying the conduct of the study and the rigorous analytical approaches, our findings and recommendations are generalizable to other low-and middle-income settings outside of SSA. It is worthy to note that, this study was unable to investigate the terminated pregnancies that were undertaken by authorized health professionals. Also, we were unable to explore the rationale for terminating these pregnancies due to the quantitative nature of the study. The relationship between RHDM and pregnancy termination is generalized for the SSA region, not the individual countries. Further studies could explore how reproductive health decisions capacity is influencing pregnancy termination in the various countries. The findings in this study can only be interpreted in terms of associations but not in causal terms. Also, there is the possibility of social desirability bias since the responses were self-reported. The answers to the main independent variable, RHDM capacity, relies mainly on a verbal report that was given by the women without validating that from their partners [21]. In spite of these, the study offers a true account of RHDM capacity and pregnancy termination in SSA. Further studies could also look at the association between socio-demographic characteristics and RHDM in SSA.

## Conclusion

We found that women who are capable of taking reproductive health decisions are more likely to terminate pregnancies. Our findings also suggest that age, level of education, contraceptive use and intention, place of residence and parity are associated with pregnancy termination. We found that women who are capable of taking reproductive health decisions are more likely to terminate pregnancies. Our findings call for the implementation of policies or the strengthening of existing ones to empower women concerning RHDM capacity. Such empowerment could have a positive impact on their uptake of safe abortions. Achieving this will not only accelerate progress towards the achievement of maternal health-related SDGs but would also immensely reduce the number of women who die as a result of pregnancy termination in SSA.

## Acknowledgments

We are grateful to Measure DHS for making the dataset available to us.

## Author Contributions

**Conceptualization:** Abdul-Aziz Seidu, Bright Opoku Ahinkorah, Edward Kwabena Ameyaw, Amu Hubert, Wonder Agbemavi, Ebenezer Kwesi Armah-Ansah, Eugene Budu, Francis Sambah, Vivian Tackie.

**Data curation:** Abdul-Aziz Seidu, Bright Opoku Ahinkorah, Edward Kwabena Ameyaw, Amu Hubert, Wonder Agbemavi, Ebenezer Kwesi Armah-Ansah, Eugene Budu, Francis Sambah, Vivian Tackie.

**Formal analysis:** Abdul-Aziz Seidu, Bright Opoku Ahinkorah, Edward Kwabena Ameyaw, Amu Hubert, Wonder Agbemavi, Ebenezer Kwesi Armah-Ansah, Eugene Budu, Francis Sambah, Vivian Tackie.

**Funding acquisition:** Abdul-Aziz Seidu, Bright Opoku Ahinkorah, Edward Kwabena Ameyaw, Amu Hubert, Wonder Agbemavi, Ebenezer Kwesi Armah-Ansah, Eugene Budu, Francis Sambah, Vivian Tackie.

**Investigation:** Abdul-Aziz Seidu, Bright Opoku Ahinkorah, Edward Kwabena Ameyaw, Amu Hubert, Wonder Agbemavi, Ebenezer Kwesi Armah-Ansah, Eugene Budu, Francis Sambah, Vivian Tackie.

**Methodology:** Abdul-Aziz Seidu, Bright Opoku Ahinkorah, Edward Kwabena Ameyaw, Amu Hubert, Wonder Agbemavi, Ebenezer Kwesi Armah-Ansah, Eugene Budu, Francis Sambah, Vivian Tackie.

**Project administration:** Abdul-Aziz Seidu, Bright Opoku Ahinkorah, Edward Kwabena Ameyaw, Amu Hubert, Wonder Agbemavi, Ebenezer Kwesi Armah-Ansah, Eugene Budu, Francis Sambah, Vivian Tackie.

**Resources:** Abdul-Aziz Seidu, Bright Opoku Ahinkorah, Edward Kwabena Ameyaw, Amu Hubert, Wonder Agbemavi, Ebenezer Kwesi Armah-Ansah, Eugene Budu, Francis Sambah, Vivian Tackie.

**Software:** Abdul-Aziz Seidu, Bright Opoku Ahinkorah, Edward Kwabena Ameyaw, Amu Hubert, Wonder Agbemavi, Ebenezer Kwesi Armah-Ansah, Eugene Budu, Francis Sambah, Vivian Tackie.

**Supervision:** Abdul-Aziz Seidu, Bright Opoku Ahinkorah, Edward Kwabena Ameyaw, Amu Hubert, Wonder Agbemavi, Ebenezer Kwesi Armah-Ansah, Eugene Budu, Francis Sambah, Vivian Tackie.

**Validation:** Abdul-Aziz Seidu, Bright Opoku Ahinkorah, Edward Kwabena Ameyaw, Amu Hubert, Wonder Agbemavi, Ebenezer Kwesi Armah-Ansah, Eugene Budu, Francis Sambah, Vivian Tackie.

**Visualization:** Abdul-Aziz Seidu, Bright Opoku Ahinkorah, Edward Kwabena Ameyaw, Amu Hubert, Wonder Agbemavi, Ebenezer Kwesi Armah-Ansah, Eugene Budu, Francis Sambah, Vivian Tackie.

**Writing – original draft:** Abdul-Aziz Seidu, Bright Opoku Ahinkorah, Edward Kwabena Ameyaw, Amu Hubert, Wonder Agbemavi, Ebenezer Kwesi Armah-Ansah, Eugene Budu, Francis Sambah, Vivian Tackie.

**Writing – review & editing:** Abdul-Aziz Seidu, Bright Opoku Ahinkorah, Edward Kwabena Ameyaw, Amu Hubert, Wonder Agbemavi, Ebenezer Kwesi Armah-Ansah, Eugene Budu, Francis Sambah, Vivian Tackie.

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
