## [Decision Letter · Decision Letter 0]

15 Apr 2020

PONE-D-19-30083

Women’s reproductive health decision-making capacity and pregnancy termination in sub-Saharan Africa: a multi-country analysis of 27 cross-sectional surveys

PLOS ONE

Dear Mr. Sambah,

Thank you for submitting your manuscript to PLOS ONE. After careful consideration, we feel that it has merit but does not fully meet PLOS ONE’s publication criteria as it currently stands. Therefore, we invite you to submit a revised version of the manuscript that addresses the points raised during the review process.

SPECIFIC ACADEMIC EDITOR COMMENTS: An expert reviewer in the field handled your manuscript. However, a number comments arose during review that require major revision to your manuscript. Please address ALL of the reviewer's comments.

We would appreciate receiving your revised manuscript by May 30 2020 11:59PM. To enhance the reproducibility of your results, we recommend that if applicable you deposit your laboratory protocols in protocols.io, where a protocol can be assigned its own identifier (DOI) such that it can be cited independently in the future. For instructions see: http://journals.plos.org/plosone/s/submission-guidelines#loc-laboratory-protocols

We look forward to receiving your revised manuscript.

Kind regards,

Frank T. Spradley

Academic Editor

PLOS ONE

"The DHS obtain ethical clearance from the Ethics Committee of ORC Macro Inc. as

well as Ethics Boards of partner organizations of the various countries such as the

Ministries of Health. During each of the surveys, either written or verbal consent was

provided by the women. Since the data was not collected by the authors of this

manuscript, we sought permission from the website of MEASURE DHS and access to

the data was provided after our intent for the request was assessed and approved on

27th January, 2019. Permission to use the data set was sought from MEASURE DHS."

Please amend your current ethics statement to confirm that your named institutional review board or ethics committee specifically approved this study.

Reviewers' comments:

Reviewer's Responses to Questions

**Comments to the Author**

1. Is the manuscript technically sound, and do the data support the conclusions?

Reviewer #1: Partly

2. Has the statistical analysis been performed appropriately and rigorously? 

Reviewer #1: I Don't Know

3. Have the authors made all data underlying the findings in their manuscript fully available?

Reviewer #1: Yes

4. Is the manuscript presented in an intelligible fashion and written in standard English?

Reviewer #1: Yes

5. Review Comments to the Author

Reviewer #1: This paper examines reproductive health decision making capacity and pregnancy termination in 27 sub-Saharan countries. The study was an informative descriptive study of the relationship between a number of factors and PT, with reproductive health decision making key among them.

Title: maybe add “and other factors related to…”. This is because the discussion is equally about RH and all other sociodemographic factors. So the paper is not solely about theory in RHDM but a more comprehensive perspective.

Abstract.

Say something about how RDM capacity is measured within the abstract.

Give an indication of which one was the dependent variable in the Binary Logistic Regression Analysis

Consistently use the term “prevalence of’ instead of prevalence then proportion in the methods vs Results

Its 27 countries in the analysis, the comparison or focus on three Gabon, Congo and Angola does not seem informative. Maybe add the associated sociodemographic factors as they were also assessed.

Conclusion: it is not clear from your results what the policy attention should be. And your study did not explore safety issues at least not mentioned in the results. Maybe your conclusions should be related more to reproductive health decision making capacity and the policy recommendation in that regard.

Background

Line 70-71: Please give a little more context. 830 women from where?

Line 73: all global UNSAFE terminations not 97% of all terminations

Line 73-76: Please provide a source.

What is the context for an unsafe PT? Is it one undertaken outside of hospital for instance? This will be important to define especially as the numbers are so high.

Line 99: Seidu et al

Line 99-106 ; Maybe link these also to then the safe PT issue. Does lack of RDM capacity also have a bearing on how safe/unsafe the abortion procured is? And also to link in objects out of the women’s control such as health facilities with no abortion services.

Line 107-109: These need to be elaborated a little more so that the reader can get a true sense of where the gap in the literature is. See also my comment above of what still needs to be discussed. If concerned about brevity, maybe cut out some of the definitions that may not be necessary to include in the beginning of the background.

Materials and Methods

127-143: I feel this can be made briefer with references to outside sources to get more detailed information on DHS sampling and data collection procedure. This is for the sake of brevity and not to lose focus on the actual materials and methods for the current study.

144-145: Is it all the women in the sample or just a sub-set of them?

170: Does capable mean two “yes” answers for each variable?

Table 1: is pregnancy terminated column a percentage?

209-210: refer to Table 2 here, so that the reader can refer to it early not at end of paragraph.

230: isn’t that 1.48, please double check.

230-232: is this difference in countries tied to the RDM capacity of women in those countries?

I wonder if an analysis of the relationship between the RDM and other socio-demographics could also be informative

Discussion

302; kindly just fix the grammar – maybe skip “our findings support the findings of’ and just put the citations at the end of the first sentence if they apply there.

Maybe list as a limitation that the relationship between RHDM and PT is generalized for the SSA region, not the individual countries.

Conclusion

349-350: Maybe clarify – PT and RHDM were linked when analyzed across countries. PT was high in Congo and Gabon yes, and in the discussion you provide a reason why. By referring to other literature. So I don’t believe this conclusion in this particular sentence is merited by your own findings. The RHDM dynamics in these individual countries or even the outcomes were not discussed specifically so I believe this conclusion is a leap. The dynamics in these countries had to do with pregnancy prevalence and lack of access to contraceptives.

The issue of safety, “medically sanctioned” and the like should be explored more in the background.

6. PLOS authors have the option to publish the peer review history of their article (what does this mean?). If published, this will include your full peer review and any attached files.

Reviewer #1: No

---

## [Author Response · Author response to Decision Letter 0]

22 Apr 2020

AUTHOR’S RESPONSE TO REVIEWS

Title: What has women’s reproductive health decision-making capacity and other factors got to do with pregnancy termination in sub-Saharan Africa? Evidence from 27 cross-sectional surveys

Dear Editor and Reviewer (s),

On behalf of all authors, I convey our gratitude to you for the critical and constructive review that has led to the improvement of our revised paper entitled “What has women’s reproductive health decision-making capacity and other factors got to do with pregnancy termination in sub-Saharan Africa? Evidence from 27 cross-sectional surveys”. We have revised the manuscript based on the comments raised. In the following detailed response, we have addressed each comment point-by-point and relevant additional texts have been added to the body of the manuscript. Most of the changes have been indicated in yellow colour. We believe the manuscript has improved substantively and will be published in your reputable journal, PLOS ONE. 

Version:1 

Manuscript ID: PONE-D-19-30083

Date: 22/04/2020

Reviewer #1: 

This paper examines reproductive health decision making capacity and pregnancy termination in 27 sub-Saharan countries. The study was an informative descriptive study of the relationship between a number of factors and PT, with reproductive health decision making key among them.

1. Comment: Title: maybe add “and other factors related to…”. This is because the discussion is equally about RH and all other sociodemographic factors. So the paper is not solely about theory in RHDM but a more comprehensive perspective.

Response: This has been revised to read “What has women’s reproductive health decision-making capacity and other factors got to do with pregnancy termination in sub-Saharan Africa? Evidence from 27 cross-sectional surveys”

Abstract.

2. Comment: Say something about how RDM capacity is measured within the abstract.

Response: We have added “Reproductive health decision-making capacity was generated from two variables: decision-making on sexual intercourse and decision-making on condom use (see page 1 line 49-53 )

3. Comment: Give an indication of which one was the dependent variable in the Binary Logistic Regression Analysis

Response: This has been done (see page 1)

4. Comment: Consistently use the term “prevalence of’ instead of prevalence then proportion in the methods vs Results

Response: Well Noted. This has been changed to “prevalence of” in the entire manuscript. 

5. Comment: Its 27 countries in the analysis, the comparison or focus on three Gabon, Congo and Angola does not seem informative. Maybe add the associated sociodemographic factors as they were also assessed.

Response: We have added significant sociodemographic factors as they were also assessed (see page 1-2 line 58-69).

6. Comment: Conclusion: it is not clear from your results what the policy attention should be. And your study did not explore safety issues at least not mentioned in the results. Maybe your conclusions should be related more to reproductive health decision making capacity and the policy recommendation in that regard.

Response: We have revised our conclusion (see page 2 line 72-81 and page 20-21 line 407-415)

Background

7. Comment: Line 70-71: Please give a little more context. 830 women from where?

Response: Done (see page 3 line 96)

8. Comment: Line 73: all global UNSAFE terminations not 97% of all terminations

Response: Done (see page 3 line 99-102)

9. Comment: Line 73-76: Please provide a source.

Response: Done (see page 3 line 102)

10. Comment: What is the context for an unsafe PT? Is it one undertaken outside of hospital for instance? This will be important to define especially as the numbers are so high.

Response: Done (see page 3-4 line 108-110).

11. Comment: Line 99: Seidu et al

Response: Done (please see page 4 line 131).

12. Comment: Line 99-106 ; Maybe link these also to then the safe PT issue. Does lack of RDM capacity also have a bearing on how safe/unsafe the abortion procured is? And also to link in objects out of the women’s control such as health facilities with no abortion services.

Response: This has been modified (see Page 5 line 134-138).

13. Comment: Line 107-109: These need to be elaborated a little more so that the reader can get a true sense of where the gap in the literature is. See also my comment above of what still needs to be discussed. If concerned about brevity, maybe cut out some of the definitions that may not be necessary to include in the beginning of the background.

Response: This has been modified (see Page 5 line 153-166)

Materials and Methods

14. Comment: 127-143: I feel this can be made briefer with references to outside sources to get more detailed information on DHS sampling and data collection procedure. This is for the sake of brevity and not to lose focus on the actual materials and methods for the current study.

Response: This has been revised to make it more concise (see Page 5 line 139-150)

15. Comment: 144-145: Is it all the women in the sample or just a sub-set of them?

Response: For this study, a sample size of 240,489 women who had complete information on reproduction health decision-making were included. Thus, women who were either married or cohabiting (living with a partner) were included. (see Page 5 line 161-163)

16. Comment: 170: Does capable mean two “yes” answers for each variable?

Response: RDM capacity, was generated by combining the decision-making on sexual intercourse and the decision-making on condom use variables. This was categorized as 0 = not capable and 1 = at least capable of taking 1 decision (see page 6 line 181-186). 

17. Comment: Table 1: is pregnancy terminated column a percentage?

Response: Yes, it was a percentage. This has been clarified (See Table 1 and 2)

18. Comment: 209-210: refer to Table 2 here, so that the reader can refer to it early not at end of paragraph.

Response: This has been done (see page 10 line 241).

19. Comment: 230: isn’t that 1.48, please double check.

Response: Please we have checked it. We reported on the AOR which was 1.20 (see page 12 line 267-269) 

20. Comment: 230-232: is this difference in countries tied to the RDM capacity of women in those countries?

Response: This was variation in countries in terms of Pregnancy terminations but not RDM. 

21. Comment: I wonder if an analysis of the relationship between the RDM and other socio-demographics could also be informative

Response: Thank you for this. We have acknowledged this as a limitation and suggest further studies should explore how socio-demographic factors influence RDM in SSA (see Page 20 line 403-404) . 

Discussion

22. Comment: 302; kindly just fix the grammar – maybe skip “our findings support the findings of’ and just put the citations at the end of the first sentence if they apply there.

Response: We have fixed these issues at the discussion section of our paper (see page…..) 

23. Comment: Maybe list as a limitation that the relationship between RHDM and PT is generalized for the SSA region, not the individual countries.

Response: Thank you. We have acknowledged this as a limitation (see page 20 line 399-340)

Conclusion

24. Comment: 349-350: Maybe clarify – PT and RHDM were linked when analyzed across countries. PT was high in Congo and Gabon yes, and in the discussion you provide a reason why. By referring to other literature. So I don’t believe this conclusion in this particular sentence is merited by your own findings. The RHDM dynamics in these individual countries or even the outcomes were not discussed specifically so I believe this conclusion is a leap. The dynamics in these countries had to do with pregnancy prevalence and lack of access to contraceptives.

Response: The conclusion has been revised (see page 20-21 line 407-415). 

25. Comment: The issue of safety, “medically sanctioned” and the like should be explored more in the background.

Response: We have added this aspect to the background (see page 3-4 line 108-110)

---

## [Decision Letter · Decision Letter 1]

15 Jun 2020

What has women’s reproductive health decision-making capacity and other factors got to do with pregnancy termination in sub-Saharan Africa? Evidence from 27 cross-sectional surveys

PONE-D-19-30083R1

Dear Dr. Sambah,

We’re pleased to inform you that your manuscript has been judged scientifically suitable for publication and will be formally accepted for publication once it meets all outstanding technical requirements.

Kind regards,

Frank T. Spradley

Academic Editor

PLOS ONE

Reviewers' comments:

Reviewer's Responses to Questions

**Comments to the Author**

1. If the authors have adequately addressed your comments raised in a previous round of review and you feel that this manuscript is now acceptable for publication, you may indicate that here to bypass the “Comments to the Author” section, enter your conflict of interest statement in the “Confidential to Editor” section, and submit your "Accept" recommendation.

Reviewer #1: All comments have been addressed

2. Is the manuscript technically sound, and do the data support the conclusions?

Reviewer #1: Yes

3. Has the statistical analysis been performed appropriately and rigorously? 

Reviewer #1: I Don't Know

4. Have the authors made all data underlying the findings in their manuscript fully available?

Reviewer #1: Yes

5. Is the manuscript presented in an intelligible fashion and written in standard English?

Reviewer #1: Yes

6. Review Comments to the Author

Reviewer #1: (No Response)

7. PLOS authors have the option to publish the peer review history of their article (what does this mean?). If published, this will include your full peer review and any attached files.

Reviewer #1: No

---

## [Editor Report · Acceptance letter]

8 Jul 2020

PONE-D-19-30083R1 

What has women’s reproductive health decision-making capacity and other factors got to do with pregnancy termination in sub-Saharan Africa? Evidence from 27 cross-sectional surveys 

Dear Dr. Sambah:

I'm pleased to inform you that your manuscript has been deemed suitable for publication in PLOS ONE. Congratulations! Your manuscript is now with our production department. 

Kind regards, 

on behalf of

Dr. Frank T. Spradley 

Academic Editor

PLOS ONE